

# The discriminative power of the ReproQ: a client experience questionnaire in maternity care

Marisja Scheerhagen[1,2], Henk F. van Stel[3,†], Arie Franx[2], Erwin Birnie[4] and Gouke J. Bonsel[5]

[1] Department of Obstetrics and Gynecology, Erasmus Medical Centre, Rotterdam, The Netherlands
[2] Department of Obstetrics and Gynecology, University Medical Center Utrecht, Utrecht, The Netherlands
[3] Department of Healthcare Innovation and Evaluation, University Medical Center Utrecht, Utrecht, The Netherlands
[4] Erasmus School of Health Policy & Management, Department of Health Technology Assessment, Erasmus University Rotterdam, Rotterdam, The Netherlands
[5] Department of Obstetrics and Gynecology, Academic Collaborative Maternity Care, University Medical Center Utrecht, Utrecht, The Netherlands
[†] Deceased.

Corresponding author
Erwin Birnie, birnie@eshpm.eur.nl

## ABSTRACT

**Background**. The aim of the ReproQuestionnaire (ReproQ) is to measure the client's experience with maternity care, following WHO's responsiveness model. To support quality improvement, ReproQ should be able to discriminate best from worst organisational units.

**Methods**. We sent questionnaires to 27,487 third-trimester pregnant women (response 31%) and to 37,230 women 6 weeks after childbirth (response 39%). For analysis we first summarized the ReproQ domain scores into three summary scores: total score (all eight domains), personal score (four personal domains), and setting score (four setting domains). Second, we estimated the proportion of variance across perinatal units attributable to the 'actual' difference across perinatal units using intraclass correlation coefficients (ICCs). Third, we assessed the ability of ReproQ to discriminate between perinatal units based on both a statistical approach using multilevel regression analyses, and a relevance approach based on the minimally important difference (MID). Finally, we compared the domain scores of the best and underperforming units.

**Results**. ICCs ranged between 0.004 and 0.025 for the summary scores, and between 0.002 and 0.125 for the individual domains. ReproQ was able to identify the best and worst performing units with both the statistical and relevance approach. The statistical approach was able to identify four underperforming units during childbirth (total score), while the relevance approach identified 10 underperforming units.

**Conclusions**. ReproQ, a valid and efficient measure of client experiences in maternity care, has the ability to discriminate well across perinatal units, and is suitable for benchmarking under routine conditions.

## INTRODUCTION

The performance of health care systems is primarily judged by health outcomes, such as mortality, morbidity, health status, or burden of disease. System performance differs across and within countries, partly caused by differences in the provision of care (*Mohangoo et al., 2011*; *Zeitlin et al., 2013a*; *Zeitlin et al., 2013b*). To highlight the role of provision of care in health system performance, the World Health Organization (WHO) introduced the measurement of client experiences with service provision and service quality as a cornerstone in health care provider evaluations (*Valentine, Bonsel & Murray, 2007*; *Valentine et al., 2003*).

Client experiences with care provision matter for at least two reasons. First, these may guide the client's choice of health care provider, particularly when the health outcomes across providers are about similar (*Valentine et al., 2003*). Second, better client experiences may contribute to improved health outcomes (*Campbell, Roland & Buetow, 2000*; *Valentine, Bonsel & Murray, 2007*; *Valentine et al., 2003*). For example, clients who understand their caregiver's explanations are more likely to comply to treatment or lifestyle changes.

To cover a broad spectrum of client experiences, independent from specific system characteristics and relevant to all medical professionals and settings, WHO developed the so-called Responsiveness concept. Responsiveness is defined as the way a client is treated by the professional and the environment in which the client is treated during a health care encounter (*De Silva, 2000*; *Gostin, 2002*; *Valentine et al., 2003*). It is based on actual performance in health practice, rather than on organisational features with claimed benefit.

Responsiveness data can be used in a universal two stage quality cycle. In the first stage, through aggregated client scores, health care providers are ranked in terms of performance. In the second stage, each underperforming organisation digs into the differences responsible for the deviant result by disaggregation of summary scores into domain scores and/or item scores. The subsequent implementation of improvement measures to target the identified weak points in service delivery should result in measurable improvement, even among average performers (*Collins-Fulea, Mohr & Tillett, 2005*). This check of improvement closes the quality cycle.

In maternity care, routine measurement of client experiences is common practice in several countries. However, to our knowledge, a fully implemented two-stage quality cycle is not yet operational in maternity care (*Dzakpasu et al., 2008*; *Hay, 2010*; *Redshaw & Heikkila, 2010a*; *Van Wagtendonk, Hoek & Wiegers, 2010*; *Wiegers et al., 1996*). A major challenge in performance measurement in maternity care, with clients being predominantly healthy young women, is the discriminative power of a measure to quantify client experiences. Poor outcomes are infrequent, and poor performance of specific client groups or units can easily be compensated by good performance of other client groups or units. Moreover, systematic variation in performance scores across units can originate from systematic variation in performance scores at the unit level even when clients are comparable across units, or from actual variation in individual performance scores of clients within units. This so-called nested or hierarchical structure of performance (units and clients within units) requires a

specific statistical approach to expose the actual performance at the perinatal unit level, but above all a measure with excellent measurement properties without becoming too lengthy or complex.

The study aim was to evaluate the discriminative power of ReproQ at the perinatal unit level (a hospital with its associated community midwife practices). ReproQ is a validated questionnaire to measure client experiences with maternity care based on WHO's Responsiveness concept (*Scheerhagen et al., 2015b*). We use two approaches to determine discriminative ability. The first, conventional, approach is to identify poor performing perinatal units on the basis of a statistically significant difference from the average performance score of all perinatal units, taking the nested nature of the data into account. In the second approach, we identified a perinatal unit as poor performer if its aggregated score deviated from the aggregated score of the best performing units by at least a minimally important or 'meaningful' difference (MID). Once discriminatory power is determined, we explore the potential for targeting the areas that need improvement. We hypothesized that ReproQ has sufficient discriminatory power for national implementation in the Dutch maternity care system, if ReproQ shows sufficient discriminatory power in both approaches, and is able to identify targets for improvement.

## MATERIALS & METHODS

### ReproQuestionnaire
ReproQ (33 items) consists of two complementary versions; the antepartum questionnaire addresses women's experiences in the first and second half of pregnancy, while the postpartum questionnaire addresses women's experiences the childbirth and the subsequent postpartum week.

The eight-domain WHO Responsiveness concept was used as the conceptual base (*Valentine, Bonsel & Murray, 2007*; *Valentine et al., 2003*). The four domains on personal interactions between the client and health professional are: dignity, autonomy, confidentiality and communication. The four domains regarding experiences with the organizational setting are: prompt attention, access to family and community support, quality of basic amenities, and choice and continuity of care.

Figure 1 shows the ReproQ scoring model. The client's responses can be summarized as (a) eight separate domain scores, (b) the personal summary score (covering the four 'personal' domains) and the setting summary score (covering the four 'setting'-related domains), and (c) the total score (covering all eight domains); a higher score implies better performance. Each score can be presented for each of the four reference periods. The summary scores of clients can be subsequently aggregated by health care provider, organisational unit, or region. Psychometric analyses support the content and construct validity as well as the test-retest reliability of the questionnaire (*Scheerhagen et al., 2015b*; *Scheerhagen et al., 2016*). For the remainder of this paper, we will only present the results of the 2nd half of pregnancy, as the ratings of the 1st and 2nd half of the pregnancy are highly associated (ICC = 0.83) (*Scheerhagen et al., 2018*).

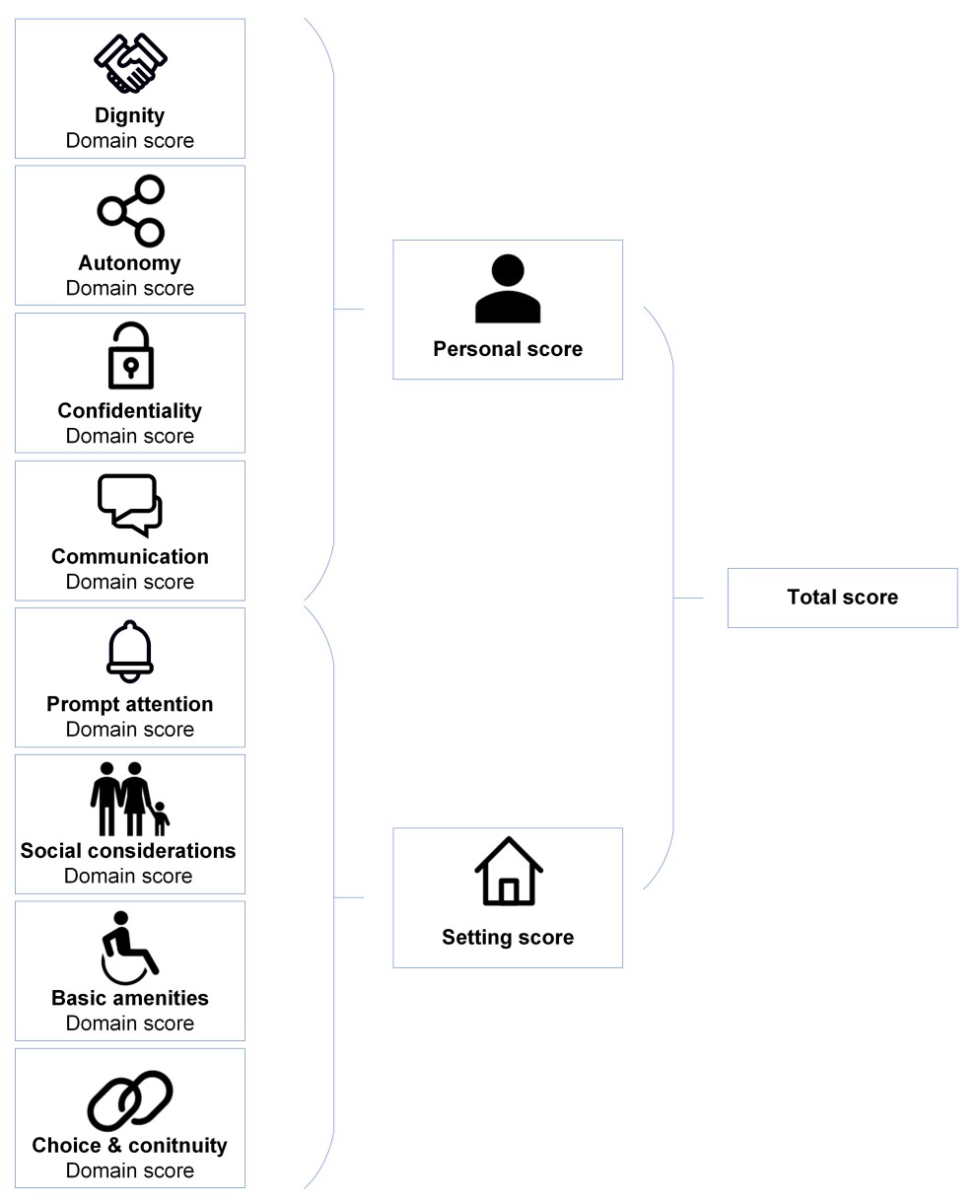

**Figure 1** ReproQ's scoring model.

## ReproQ data collection

The ReproQ data were collected from August 2013 to July 2015. The data were collected digitally from clients of three large maternity care organizations (that deliver postnatal care at home from childbirth onwards during 7–10 days), from several regionally conducted observational studies that used ReproQ client experiences as secondary outcome measure, and from clients of 10 perinatal units interested in quality improvement. A total of 60 of 85 Dutch perinatal units participated. Further details have been reported elsewhere (*Scheerhagen et al., 2016*).

There were no exclusion criteria. All women could participate provided that they gave informed consent. Invitations to fill out the antepartum ReproQ were sent around 34th weeks' gestational age. The postpartum ReproQ was sent 6 weeks after the expected date of childbirth. Non-responding clients received a reminder two weeks after the initial invitation. The study protocol and procedures were approved by the Medical Research Ethics Review Board, Erasmus Medical Center, Rotterdam, the Netherlands (MEC-2013-455).

### Unit of analysis: perinatal unit

The perinatal unit was the unit of analysis. At the time the study was conducted, each perinatal unit contained one and only one hospital. Consequently, clients were allocated to the hospital's perinatal unit in case of a hospital birth. In case of an out-of-hospital birth, clients were allocated to the hospital (and perinatal unit) that was closest to the client's home address. Descriptive characteristics of perinatal units were obtained from various public sources (*Posthumus et al., 2015*; *STZ Foundation, 2014*; *PRN Foundation, 2013*).

### Excluded data

Excluded from analysis were: (1) ReproQ data of clients who could not be allocated to one perinatal unit (72 clients, 0.7%); (2) ReproQ data with >50% missing answers in two or more ReproQ domains; and (3) data of perinatal units who included less than 50 clients.

### Analytical framework: multilevel analysis

Crude differences in summary ReproQ scores across perinatal units, the dependent variable, may originate from three sources: (1) 'actual' differences across perinatal units, (2) differences in client characteristics across perinatal units, and (3) residual variance. Given the hierarchical data structure (perinatal units, and clients within perinatal units), existing differences in client characteristics across perinatal units may obscure the estimation of the 'actual' difference across units. In that case, multi-level analysis is the appropriate method to decompose total data variance into variance attributable to perinatal units (source 1) and variance attributable to other sources (sources 2+3), in particular variance related to client characteristics (*Twisk, 2014*). Estimation of the 'actual' difference between perinatal units (source 1) requires the domain and summary scores to be corrected for the other variance components (typically client characteristics), as systematic client heterogeneity may bias and limit the comparison of perinatal units; i.e., case mix correction. The Technical Supplement shows further details on the multilevel analysis and software used.

## Casemix correction

For a fair ranking of units, we explored the need for case mix correction. We distinguished the following groups of potential casemix variables: (1) socio-demographic variables; (2) process of care variables; (3) interventions; and (4) perceived (client-reported) outcomes. The variables in the casemix correction were limited to variables that cannot be modified by perinatal units and healthcare providers, i.e., socio-demographic variables and perceived (patient reported) outcomes.

To explore the need for casemix correction, we analyzed two models: (1) an 'empty' model with the ReproQ domain or summary scores as dependent variable, and a random

intercept for each perinatal unit; (2) an adjusted model, with the ReproQ domain or summary score as dependent variable, a random intercept for each perinatal unit, and client characteristics included as explanatory variables (*Bos et al., 2015*; *Stubbe, Brouwer & Delnoij, 2007*). Of the available client characteristics (age, educational level, ethnicity, parity, and client-reported health) only age, educational level, and client-reported health contributed significantly ($p < 0.05$) to all domain and summary scores, and were therefore included in the casemix correction in all analyses. We also tested for random slopes, but none of these were significant and therefore remain unreported.

## Discriminative ability: two approaches

We used two complementary approaches to determine discriminative power.

### *Approach 1. Multilevel testing of the deviation of unit means from overall (grand) mean*

Multi-level analyses were used to examine to what degree ReproQ is able to identify units that significantly perform above and below average (averaged over perinatal units), producing three parts of information: (1) Estimated variance components and ICCs. The ICC of interest is the ratio of the variance in perinatal units and the variance in client's characteristics in that unit (*Streiner, Norman & Cairney, 2014*). An ICC close to zero implies that the client's experience is unrelated to the perinatal unit in which one receives care; an ICC close to one means that the perinatal unit is of decisive importance. Best and poor performing units are identified by the deviation of the 95% CI of each individual perinatal unit from the grand mean of all perinatal units. (2) Estimated G-coefficients, which represent the proportion of variance in the unit-level mean scores attributable to 'actual' variation among perinatal units. A G-coefficient of one implies that all variance in domain and summary scores across perinatal units is attributed to the perinatal unit, and no variance can be attributed to other sources. (3) Estimation of the minimal number of clients needed per perinatal unit to achieve sufficient reliability (D-Study), in our study defined as 0.80. Small numbers of clients and large heterogeneity in client experiences produce wide confidence intervals, but only the numbers of clients can be influenced.

The conventional mode of presentation is the so-called caterpillar-plot; see Fig. 2.

### *Approach 2. Relevant deviation based on MID*

This approach judges ReproQ's discriminative power on the basis of the ability to demonstrate a relevant difference in domain or summary scores at the perinatal unit level, defined as a difference score beyond the so-called minimally important difference (MID). Underperforming units are identified as those units with domain or summary scores below a certain threshold that equals the mean score domain or summary score of the 10% best performing units minus the MID. We previously determined, at the individual level, the minimally important difference (MID) for both the summary and domain scores of the childbirth period (*Scheerhagen et al., 2016*). For the study reported here, we derived MIDs for the late pregnancy and postnatal reference periods in a similar way. Note that a difference of 1.0 unit of MID at the perinatal unit level reflects a large difference: it means that *all* clients cared for in that unit, on average differ 1.0 MID from a reference value,
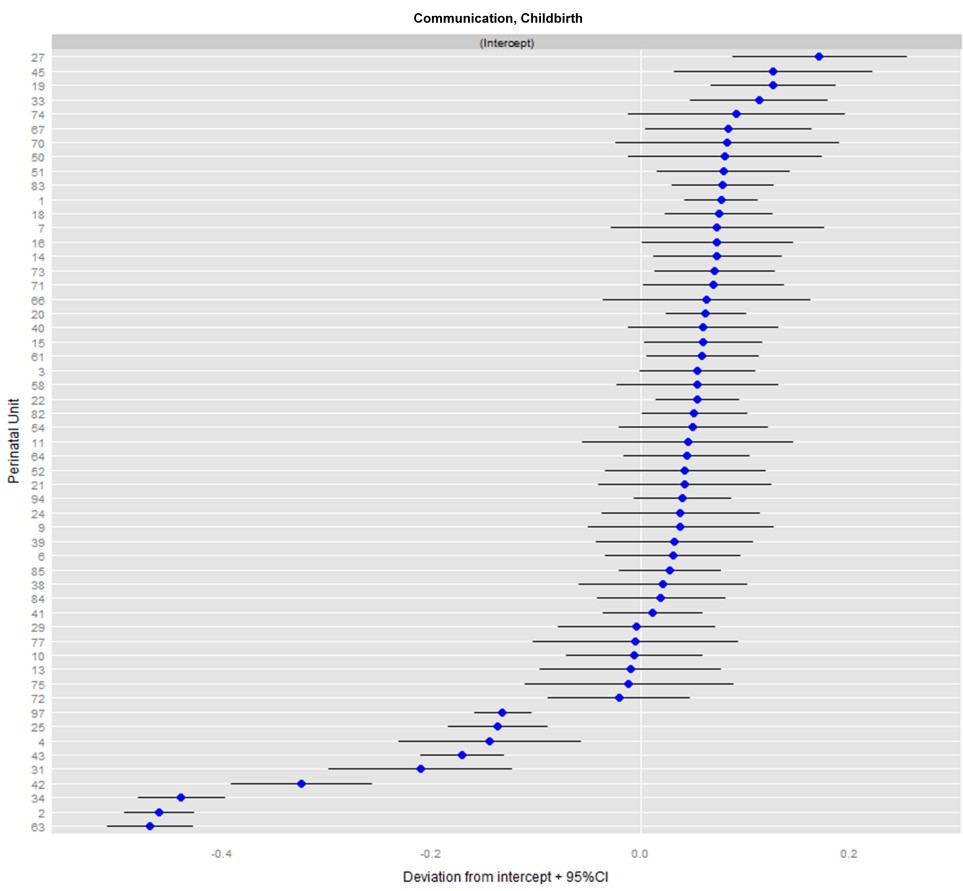

**Figure 2** Caterpillar-plot: ranking of perinatal units for the domain communication during childbirth ($N_{PU} = 55$).

either being much better (best practice) or much worse (poor practice). We also presented results for a more conservative 0.5 MID.

### Profiling underperforming units

Assuming sufficient discriminatory power, we explored ReproQ's ability to profile underperforming perinatal units, once these units have been identified. For that purpose, we compared the domain scores (reference period: childbirth) of the statistically best performing perinatal unit with the significantly underperforming perinatal units. Subsequently, once the underperforming domains have been identified, profiling can be applied to the items of the domains. A detailed comparison of the items may provide clues to the precise activity (or activities) where underperforming units should improve. We illustrate this stepwise approach of underperformance to guide quality improvement using the Communication-domain.

## RESULTS

We invited 27,487 pregnant women to participate in the antepartum ReproQ (response: 8,567 (31%)) and 37,230 women who recently had given birth to respond to the postpartum ReproQ (response: 12,477 (39%)). Excluded from analysis were 1,419 pregnant women and 1,751 women who recently had given birth, for having >50% missing answers. Additionally, we excluded 761 pregnant women and 1,080 women who recently had given birth, for whom the perinatal unit code was missing or being a perinatal unit with less than 50 responses.

Table 1 shows the characteristics of participating clients and perinatal units. Differences between the antepartum and postpartum client and unit characteristics were minimal, and about representative for the national pregnancy population.

Table 2 presents the results of the multi-level analysis of the corrected model which includes perinatal unit as random effect and the client characteristics age, educational level and client-reported health as case mix correction variables. Each row represents a separate analysis for the experience measure shown (total score, domain score) in each of the three periods (pregnancy, childbirth, postnatally). Columns #2-4 provide the estimated ICCs, or the ratio of variance assigned to perinatal units (column #2) and variance assigned to client characteristics (column #3). For example, the first row shows the results for the antepartum total ReproQ score. It shows that little variance on the client level can be assigned to the perinatal units in general (0.0001), and somewhat more to the variance of client characteristics (0.064). The ICC is 0.011 (column #4), indicating that the client's experience is to a limited extent related to the perinatal unit in which one received care. Column #5 shows that the G-coefficients (or proportion of variance in the mean scores of perinatal units that can be attributed to the 'actual' variation among perinatal units; the higher the better) of the total score during pregnancy is 0.63. Finally, the 6th column shows that at least 272 respondents per unit are needed to achieve a reliability (G-coefficient) of 0.80.

As Table 2 shows, all ICCs of the three summary scores for all three reference periods were lower than 0.03. Moreover, the ICCs of the case mix corrected models range from 0.002 (the domains dignity, confidentiality and choice & continuity during childbirth) to 0.125 (communication during childbirth). Moreover, the ICCs for the individual domains showed more variability than the summary scores, with Communication showing the highest ICC.

The G-coefficients of the summary scores ranged from 0.45 (personal score during pregnancy) to 0.75 (setting score during postnatal period). The G-coefficients of the domain scores ranged between 0.19 (dignity during childbirth) to 0.93 and 0.96 (communication during childbirth and postnatal period). For the antepartum period, prompt attention (0.74) and basic amenities (0.70), both part of the setting score, were the domains with highest G-coefficients. The number of respondents needed to achieve a G-coefficient of 0.80 ranged from 18 (Communication during Childbirth) to 1,910 (Dignity during Childbirth). The total scores of the ReproQ would achieve excellent reliability (G-coefficient of 0.80)
**Table 1** **Characteristics of the participating women ($n_{antepartum} = 6,387$; $n_{postpartum} = 9,646$) and perinatal units ($n_{antepartum} = 42$; $n_{postpartum} = 55$).** (A) Mean age was 30.1 years ($SD = 4.5$). (B) Educational level; low 0–6 years; middle 6–12 years; high > 12 years. (C) Mean number of respondents per perinatal unit was 152 (range: 54–363) for the antenatal period, and 175 (range: 50–812) for the postnatal period.

| | Antapartum questionnaire | | Postpartum questionnaire | |
|---|---|---|---|---|
| | **N** | **%** | **N** | **%** |
| **Clients** | | | | |
| Age (A) | | | | |
| ≤24 | 385 | 6 | 500 | 5 |
| 25–29 | 2,018 | 32 | 2,730 | 29 |
| 30–34 | 2,600 | 42 | 4,084 | 43 |
| ≥35 | 1,263 | 20 | 2,197 | 23 |
| Ethnic background | | | | |
| Western | 5,735 | 92 | 8,711 | 93 |
| Non-Western | 478 | 8 | 696 | 7 |
| Educational level (B) | | | | |
| Low | 399 | 6 | 754 | 8 |
| Middle | 2,026 | 33 | 3,280 | 35 |
| High | 3,783 | 61 | 5,356 | 57 |
| Marital status | | | | |
| Married/living together | 5,974 | 96 | 9,052 | 96 |
| Not living together or no relationship | 226 | 4 | 339 | 4 |
| Parity | | | | |
| Primiparous | 3,210 | 50 | 4,872 | 51 |
| Multiparous | 3,153 | 50 | 4,735 | 49 |
| Health status | | | | |
| Poor / moderate | 300 | 5 | 332 | 4 |
| Good | 2,173 | 36 | 3,153 | 33 |
| Very good | 2,390 | 39 | 3,684 | 38 |
| Excellent | 1,244 | 20 | 2,428 | 25 |
| **Perinatal units** | | | | |
| Number of respondents (C) | | | | |
| 50–99 | 16 | 38 | 19 | 35 |
| 100–149 | 10 | 24 | 14 | 25 |
| 150–199 | 6 | 14 | 7 | 13 |
| ≥200 | 10 | 24 | 15 | 27 |
| Urbanization | | | | |
| Urban - 4 largest cities | 10 | 24 | 14 | 25 |
| Urban - 10 largest cities, except no. 1-4 | 6 | 14 | 6 | 11 |
| Rural | 26 | 62 | 35 | 64 |
| Hopsital type | | | | |
| University hospital | 5 | 12 | 6 | 11 |
| Teaching hospital | 17 | 40 | 20 | 36 |
| Peripheral hospital | 20 | 48 | 29 | 53 |
| Hospital size | | | | |
| <750 births a year | 5 | 12 | 6 | 11 |
| 750–1499 births a year | 20 | 48 | 26 | 47 |
| ≥1,500 births a year | 17 | 40 | 23 | 42 |

**Table 2  Results of corrected multi-level analysis model and G-study for ReproQ summary scores and domain scores during pregnancy ($n = 6,387$), and childbirth and postnatal period ($n = 9,646$).** (A) Mean valid response per perinatal unit was 116 antenatally and 109 postnatally.

|  | Variance of perinatal units | Variance of client characteristics | ICC | G-coefficient (A) | Number of respondents needed for G-coefficient of 0.8 |
|---|---|---|---|---|---|
| **Pregnancy** | | | | | |
| Total score | 0.001 | 0.064 | 0.011 | 0.63 | 272 |
| Personal score | 0.000 | 0.071 | 0.004 | 0.45 | 580 |
| Setting score | 0.001 | 0.082 | 0.014 | 0.66 | 243 |
| Dignity | 0.000 | 0.073 | 0.004 | 0.38 | 745 |
| Autonomy | 0.001 | 0.190 | 0.006 | 0.45 | 555 |
| Confidentiality | 0.001 | 0.204 | 0.005 | 0.38 | 755 |
| Communication | 0.000 | 0.115 | 0.004 | 0.35 | 875 |
| Prompt attention | 0.002 | 0.104 | 0.021 | 0.74 | 165 |
| Social considerations | 0.001 | 0.154 | 0.004 | 0.35 | 825 |
| Basic amenities | 0.002 | 0.111 | 0.019 | 0.70 | 202 |
| Choice and continuity | 0.001 | 0.271 | 0.003 | 0.42 | 630 |
| **Childbirth** | | | | | |
| Total score | 0.001 | 0.075 | 0.009 | 0.50 | 432 |
| Personal score | 0.002 | 0.101 | 0.019 | 0.71 | 176 |
| Setting score | 0.001 | 0.072 | 0.008 | 0.49 | 465 |
| Dignity | 0.000 | 0.115 | 0.002 | 0.19 | 1,910 |
| Autonomy | 0.007 | 0.333 | 0.020 | 0.66 | 210 |
| Confidentiality | 0.001 | 0.214 | 0.002 | 0.23 | 1,465 |
| Communication | 0.021 | 0.147 | 0.125 | 0.96 | 18 |
| Prompt attention | 0.001 | 0.123 | 0.009 | 0.46 | 523 |
| Social considerations | 0.000 | 0.105 | 0.003 | 0.23 | 1,480 |
| Basic amenities | 0.002 | 0.080 | 0.023 | 0.73 | 165 |
| Choice and continuity | 0.001 | 0.260 | 0.002 | 0.23 | 1,440 |
| **Postnatal period** | | | | | |
| Total score | 0.001 | 0.080 | 0.015 | 0.63 | 258 |
| Personal score | 0.001 | 0.100 | 0.012 | 0.58 | 320 |
| Setting score | 0.002 | 0.081 | 0.025 | 0.75 | 145 |
| Dignity | 0.001 | 0.122 | 0.009 | 0.51 | 425 |
| Autonomy | 0.001 | 0.287 | 0.004 | 0.30 | 1,015 |
| Confidentiality | 0.001 | 0.212 | 0.007 | 0.42 | 600 |
| Communication | 0.011 | 0.145 | 0.073 | 0.93 | 35 |
| Prompt attention | 0.002 | 0.096 | 0.018 | 0.66 | 221 |
| Social considerations | 0.001 | 0.126 | 0.008 | 0.51 | 418 |
| Basic amenities | 0.004 | 0.108 | 0.035 | 0.78 | 123 |
| Choice and continuity | 0.003 | 0.268 | 0.013 | 0.64 | 249 |

when all perinatal units would have included 272 (antepartum), 432 (childbirth), and 258 (postnatal period) valid responses.

Figure 2 shows the caterpillar-plot for the communication domain during childbirth. Depicted are the corrected means (and 95% CIs) of all 55 perinatal units, which allows

for comparison with the grand mean of all perinatal units. The varying CI widths point to heterogeneity (after case mix correction) and different sample sizes per unit. For example, unit #22 performs only moderately better, and does so significantly, due to its small dispersion.

Table 3 presents the discriminative power according to the statistical and relevance-based approaches. Using the total score during pregnancy (1st row) discriminative power according to the statistical approach would imply that three perinatal units showed a significantly better total score compared to the grand mean (column #3), 38 units where about average (column #4), and one unit showed a below-average score (column #5). For the total score, discriminative power using the statistical approach was largest for the postnatal period (10/55 units being deviant), followed by childbirth (5/55 being deviant) and the antepartum period (4/38 being deviant). Of the summary scores, only the personal score in the antenatal period did not statistically discriminate. Overall, the domains communication and basic amenities during childbirth (with 23/55 and 16/55 units, respectively, being deviant) and during the postnatal period (23/55 and 15/55, respectively) were the domains that discriminated best, due its high reliability (see Table 2).

Table 3 columns #6-9 reveal ReproQ's discriminative power based on the MID. For the total score during pregnancy (1st row), the 10% best performing units have a mean total score of 3.80 (reference value). The corresponding MID is 0.11. Applying this MID of 0.11 implies that seven perinatal units with their CI perform below this reference. For the summary scores, the number of perinatal units that differed more than 1.0 unit of MID compared to the reference value ranged from seven (both the total and personal scores in pregnancy) to 29 (setting score in the postnatal period). The domains with most discriminating power differed for the three reference periods: autonomy and basic amenities during pregnancy, basic amenities and social considerations during childbirth, and communication, social considerations and choice & continuity during postnatal period (see column #8). Applying a conservative 0.5 unit of MID considerably increased the number of units that relevantly deviated from reference for all scores and reference periods.

Figures 3 and 4 illustrate the potential of ReproQ to profile units being selected as poor performer. Figure 3 displays the domain scores of the single best and four worst performing perinatal units, with best and underperformance defined according to the total score during childbirth. Poor performing units showed systematically lower scores for all domains.

Figure 4 displays the profiles of the four underperforming perinatal units by disaggregating the Communication domain score into its item scores. The low Communication domain score was predominantly associated with one specific item: 'giving consistent advice'.

## DISCUSSION

ReproQ, a validated instrument for measuring service quality in maternity care, has the ability to discriminate well across perinatal units using two complementary approaches; a multilevel significance-based approach and a relevance-based analysis (MID). It did
**Table 3  Discriminative power of the ReproQ based on statistical power (significance perspective) and the ability to detect 1.0 and 0.5 MID difference (relevance perspective) for all ReproQ outcomes during pregnancy ($n_{pu}$ = 42), childbirth and postnatal period ($n_{pu}$ = 55). (A) The mean best-practices is the pooled average of the 10% best performing units.**

| | Overall mean | Discriminative power based on statistics | | | Discriminative power based on relevance | | | |
| --- | --- | --- | --- | --- | --- | --- | --- | --- |
| | | Best-practices | Average | Under performers | Mean best practices (A) (ΔP90 - P100) | MID | Under performers (ΔP90 1MID) | Under performers (ΔP90 0.5MID) |
| **Pregnancy** | | | | | | | | |
| Total score | 3.73 | 3 | 38 | 1 | 3.80 | 0.11 | 7 | 22 |
| Personal score | 3.75 | 0 | 42 | 0 | 3.81 | 0.09 | 7 | 28 |
| Setting score | 3.72 | 3 | 35 | 4 | 3.79 | 0.12 | 9 | 23 |
| Dignity | 3.87 | 0 | 42 | 0 | 3.91 | 0.07 | 10 | 25 |
| Autonomy | 3.65 | 1 | 40 | 1 | 3.75 | 0.11 | 21 | 34 |
| Confidentiality | 3.73 | 0 | 41 | 1 | 3.81 | 0.09 | 12 | 36 |
| Communication | 3.75 | 0 | 42 | 0 | 3.80 | 0.11 | 6 | 21 |
| Prompt attention | 3.67 | 5 | 32 | 5 | 3.77 | 0.11 | 19 | 30 |
| Social considerations | 3.76 | 0 | 42 | 0 | 3.84 | 0.09 | 16 | 30 |
| Basic amenities | 3.81 | 4 | 34 | 4 | 3.89 | 0.08 | 22 | 29 |
| Choice and continuity | 3.63 | 0 | 42 | 0 | 3.74 | 0.19 | 7 | 26 |
| **Childbirth** | | | | | | | | |
| Total score | 3.73 | 1 | 50 | 4 | 3.80 | 0.10 | 10 | 40 |
| Personal score | 3.66 | 5 | 46 | 4 | 3.75 | 0.11 | 13 | 39 |
| Setting score | 3.79 | 4 | 50 | 1 | 3.86 | 0.08 | 21 | 44 |
| Dignity | 3.83 | 1 | 54 | 0 | 3.88 | 0.09 | 9 | 35 |
| Autonomy | 3.44 | 7 | 42 | 6 | 3.61 | 0.17 | 25 | 43 |
| Confidentiality | 3.77 | 0 | 55 | 0 | 3.84 | 0.08 | 32 | 49 |
| Communication | 3.62 | 15 | 32 | 9 | 3.80 | 0.11 | 17 | 44 |
| Prompt attention | 3.77 | 1 | 51 | 3 | 3.84 | 0.10 | 18 | 42 |
| Social considerations | 3.86 | 0 | 55 | 0 | 3.92 | 0.04 | 42 | 50 |
| Basic amenities | 3.88 | 7 | 39 | 9 | 3.94 | 0.05 | 36 | 48 |
| Choice and continuity | 3.66 | 1 | 54 | 0 | 3.77 | 0.13 | 23 | 44 |
| **Postnatal period** | | | | | | | | |
| Total score | 3.74 | 5 | 45 | 5 | 3.83 | 0.10 | 18 | 45 |
| Personal score | 3.71 | 4 | 47 | 4 | 3.80 | 0.10 | 19 | 46 |
| Setting score | 3.78 | 8 | 39 | 8 | 3.86 | 0.10 | 29 | 42 |
| Dignity | 3.81 | 4 | 48 | 3 | 3.90 | 0.10 | 21 | 45 |
| Autonomy | 3.71 | 3 | 52 | 0 | 3.82 | 0.12 | 31 | 47 |
| Confidentiality | 3.74 | 3 | 50 | 2 | 3.82 | 0.11 | 23 | 41 |
| Communication | 3.59 | 13 | 32 | 10 | 3.75 | 0.09 | 33 | 47 |
| Prompt attention | 3.80 | 6 | 42 | 7 | 3.87 | 0.09 | 21 | 41 |
| Social considerations | 3.81 | 5 | 49 | 1 | 3.89 | 0.07 | 33 | 48 |
| Basic amenities | 3.86 | 7 | 40 | 8 | 3.95 | 0.08 | 27 | 44 |
| Choice and continuity | 3.65 | 5 | 49 | 1 | 3.79 | 0.14 | 34 | 48 |

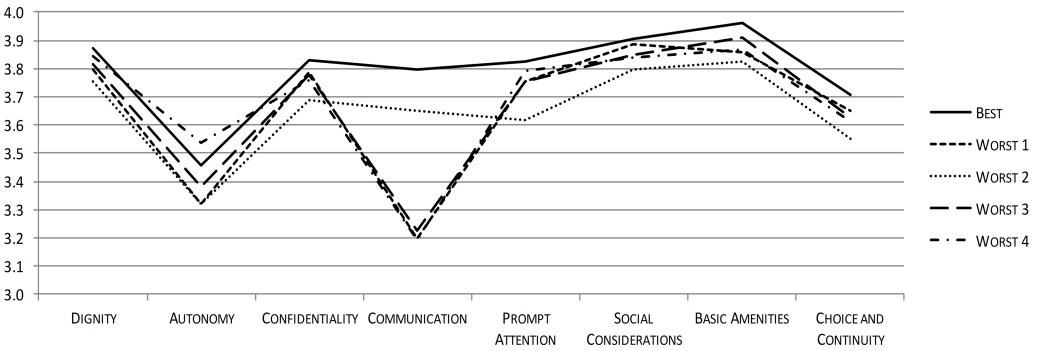

**Figure 3   ReproQ domain scores of the single best practice and the four worst performing units during childbirth.**

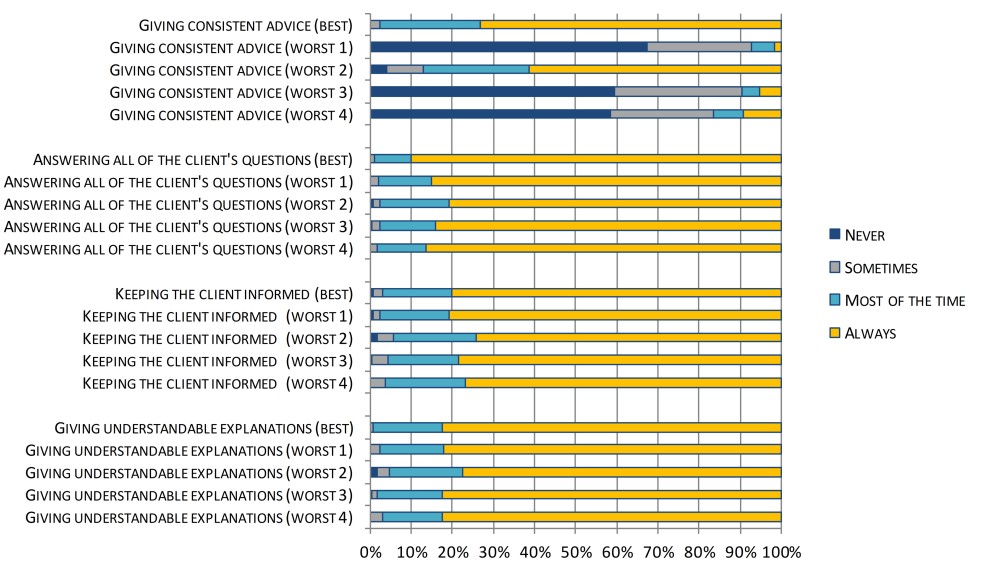

**Figure 4   ReproQ item scores of the Communication-domain of the single best practice and the four worst performing units during childbirth.**

so successfully despite four conditions that could limit its discriminative performance: a predominantly healthy and relatively homogenous population, standardized care procedures, a naturalistic study design, and the use of aggregated means. Using the total score during childbirth, the significance approach identified four underperforming units, whereas the MID-based approach identified 10 underperforming units using a 10% best-practice norm and 1.0 MID as cut-off. Once the underperforming units were identified, ReproQ domain and item scores provided useful disaggregated information for quality improvement.

A study strength is that sample size was large and data were collected in routine practice, covering about 2/3s of perinatal units. Clients and practices covered the full range of relevant characteristics, adding to generalizability. Secondly, the significance-based and

relevance-based approaches yielded consistent results, with the latter approach displaying considerably more sensitivity. The significance-based approach distinguishes between the observed outcome distribution (here: unit means) and simply tests if 'unit' has a significant impact on outcomes. Significance relies on the number of respondents, case mix correction, and details of the multilevel analysis, with the intrinsic risk that homogeneity of clients and units can lead to significant results, even when these differences are not meaningful. The reverse is more common: client heterogeneity within units and measurement error can obscure relevant unit differences. The relevance-based approach inevitably relies on the chosen reference (i.e., 10% best units) and magnitude of the MID (i.e., 0.5 or 1.0 MID as threshold). One should be aware that all our choices in both approaches were rather conservative, and that an *average* difference of 1.0 MID at the unit level expresses a rather extreme difference. Other studies on client experiences only explored the discriminative power in statistical terms (*Bos et al., 2015*; *De Boer, Delnoij & Rademakers, 2011*; *Krol et al., 2015*; *Stubbe, Brouwer & Delnoij, 2007*). We believe the MID-based approach is a necessary complement to the significance-based approach.

A third strength is that we avoided overfitting and overcorrection by limiting case mix correction to predefined candidate factors with an accepted established effect (*Sixma et al., 2008*).

Two limitations merit discussion. Firstly, while the respondents were largely representative for the pregnancy population, non-Western women were somewhat underrepresented (8% vs. national average 14% (*PRN Foundation, 2013*)). Since non-Western women tend to report more negative experiences than Western women (*Scheerhagen et al., 2015a*), increased participation of non-Western women would probably lower the average summary scores but not affect the ranking of perinatal units, as our case mix analysis did not reveal a significant role of ethnicity. Regarding age, women younger than 24 years were slightly underrepresented (6% vs 11%), as were women with a low educational level (6–8% vs. 18%). These gaps appear modest in quantitative terms. More important, a study we conducted recently on the determinants of experience scores showed that maternal age and educational level have no significant impact on the level of experience scores (*Scheerhagen et al., 2019*). Reference data on client-reported health status are absent.

Secondly, we did not include the individual professional as additional level in the analysis. One may assume an effect of individual professional's behaviour on the personal domains rather than the setting domains, and its impact is probably larger than the variation across units (*Krol et al., 2015*; *Roberts et al., 2014*). While the primary focus of quality improvement is the unit, one should be aware of the professional's role in quality improvement cycles. Inclusion of the professional in the analysis would require a highly detailed, perhaps unfeasible, registration of all caregivers involved the care process.

Three technical remarks can be made. First, a study of performance at the domain level requires about 450 respondents per unit, which is considerably higher than the minimum of 10 respondents per unit adopted in similar studies (*Bos et al., 2015*; *Stubbe, Brouwer & Delnoij, 2007*). The view that 10 respondents are representative for a unit's performance is highly questionable, given the variability in respondent characteristics, in experiences within a unit, and in the care provided. A sample size of 450 clients is well below the

average unit size of 2,000 clients, implying that a sampling approach instead of all-client measurement should suffice.

Secondly, although the estimated ICCs appeared low, they are comparable to the ICCs of other accepted client experience questionnaires (*Bos et al., 2015*; *De Boer, Delnoij & Rademakers, 2011*; *Krol et al., 2015*; *Stubbe, Brouwer & Delnoij, 2007*). ICCs are low because the denominator essentially is the number of client questionnaires. The impact of each unit on each individual questionnaire is small. Small effects at the client level may build up as large effects at the unit level.

Finally, the systematic effect of perinatal unit was stronger during childbirth and postnatally than antenatally. The likely explanation is that antenatal care is highly standardized in terms of procedures and professionals involved whereas different processes, adverse events and outcomes do emerge during childbirth and postnatally, where unit quality is challenged. This observation emphasizes that the very assumption that quality differs across units may not be true when care is highly standardized. In that case, differences across units truly are small, causing low ICCs and lack of discrimination. We believe that favorable antenatal performance should be interpreted primarily as uniform performance rather than good performance (*Scheerhagen et al., 2015b*; *Scheerhagen et al., 2016*). This phenomenon has been described with other client experience questionnaires as well (*Peterson et al., 2005*; *Redshaw & Heikkila, 2010b*; *Smith, 2011*).

Once underperforming units have been identified, profiling of its items (which are described in terms of activities) may guide interventions to improve service quality. Qualitative interviews and client involvement may further support the interpretation of domain and item scores. Also research into the varying performance of client subgroups, e.g., deprived clients, is recommended (*Department of Health, 2010*; *Ellis, 2006*; *Ettorchi-Tardy, Levif & Michel, 2012*; *Hitzert et al., 2016*; *Kay, 2007*). After identifying areas that need improvement, experts could be consulted to inform on the causes of suboptimal ReproQ domain and item scores and recommend actions for improvement. Here, both clients and health care professionals can take up the role of experts (*Groenen et al., 2017*). Another strategy is to derive recommendations for quality improvement from multidisciplinary meetings or audits, a strategy occasionally used in maternity care (*Alderliesten et al., 2008*; *Eskes et al., 2014*; *Kurinczuk et al., 2014*; *Mancey-Jones & Brugha, 1997*).

For the future, we recommend using ReproQ in maternity care to measure clients' experiences with care and using the results to guide the improvement of the performance in maternity care by means of profiling. Qualitative interview, client involvement and audits may further support this process. This may fit well in outcome-based strategies like those initiated by *ICHOM (2016)*, that includes both medical outcomes and quality of care.

## CONCLUSION

ReproQ, a valid and efficient measure of client experiences in maternity care, has the ability to discriminate well across perinatal units, and is suitable for benchmarking under routine conditions.

## ACKNOWLEDGEMENTS

We are grateful to Koen Ilja Nijenhuijs for his assistance with the caterpillar plots.

### Funding

This article was funded by the Miletus Foundation in The Netherlands. The funders had no role in study design, data collection and analysis, decision to publish, or preparation of the manuscript.

### Grant Disclosures

The following grant information was disclosed by the authors:
Miletus Foundation in The Netherlands.

### Competing Interests

The authors declare there are no competing interests.

### Author Contributions

- Marisja Scheerhagen, Henk F. van Stel and Erwin Birnie conceived and designed the experiments, performed the experiments, analyzed the data, contributed reagents/materials/analysis tools, prepared figures and/or tables, authored or reviewed drafts of the paper, approved the final draft.
- Arie Franx conceived and designed the experiments, authored or reviewed drafts of the paper, approved the final draft.
- Gouke J. Bonsel conceived and designed the experiments, contributed reagents/materials/analysis tools, authored or reviewed drafts of the paper, approved the final draft.

### Human Ethics

The following information was supplied relating to ethical approvals (i.e., approving body and any reference numbers):

The study protocol and procedures were approved by the Medical Research Ethics Review Board, Erasmus Medical Center, Rotterdam, the Netherlands (MEC-2013-455).

### Data Availability

The raw data is available in the Supplemental Files.

### Supplemental Information

Supplemental information for this article can be found online at http://dx.doi.org/10.7717/peerj.7575#supplemental-information.

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
