# Peer review of "The discriminative power of the ReproQ: a client experience questionnaire in maternity care"

_PeerJ, doi:10.7717/peerj.7575_

## Round 0.1 · original submission · Minor Revisions

This manuscript is well articulated.

For further improvement, following points have to be addressed:

Line 70-76 – Please add some connecting text between this paragraph and previous one
Line 86-87- simplify it. Difficult to comprehend
Line 150-151- Clients who could not be assigned to one perinatal unit were excluded from analysis. It would be interesting to know whether these clients switched hospitals or providers and reasons thereof. Reasons might be related to personal and facility level domains in ReproQ. Or whether non-assignment was due to mere non-availability of data.

Regarding responses to reviewer comments, you may skip to address following points

Reviewer-1
"For line 112-121- I suggest to add details in a tabular manner for better understanding." Here instead of Table you can create Figure to summarize ReproQ domains and scoring. You may consider this as optional for revision.
No need to address "Writing column variable name rather than writing column number will enhance understanding of the readers for example line 236-237.
As per Table 1: 6% antenatal women and 8% postpartum women participated from low education level. The possible impact of low education on study findings can be discussed in ‘discussion section’.".

Reviewer-2
"Since we expect women who experienced complications may experience more of negative picture and complications are commonly occur during childbirth and postpartum attempt for sensitivity analysis will ensure the robustness of the findings." With this comment, whether presence of complication was included in Case-mix correction as explanatory variable?

Rest of the comments of reviewers may please be addressed.

In case of any further clarification, please write to us.

Reviewer 1 ·

Basic reporting

The paper needs an edit since few word are either missing or needs replacement like line 47- "mortality" repeated, line 57-rewrite sentence.
I suggest adding few lines about relevant prior literature.
Add for reference line 108: Previous results indicate that ratings of the 1st half of pregnancy remain highly associated (ICC=0.80) with ....."
For line 112-121- I suggest to add details in a tabular manner for better understanding.
Line 129-132- mention duration of data collection

Experimental design

Study design is well explained.
No comments

Validity of the findings

Study findings are explained in detail and adequate for the research topic. Writing column variable name rather than writing column number will enhance understanding of the readers for example line 236-237.
As per Table 1: 6% antenatal women and 8% postpartum women participated from low education level. The possible impact of low education on study findings can be discussed in ‘discussion section’.

Reviewer 2 ·

Basic reporting

Statistically, this has been very well approached and vividly documented
However, while reading it gives a feeling there is more of statistical focus.
Adding quality improvement and challenges in assessing the quality in reproductive care, what has been conventionally followed, what are the lacunae of those approaches might add value to this article

Experimental design

Since we expect women who experienced complications may experience more of negative picture and complications are commonly occur during childbirth and postpartum attempt for sensitivity analysis will ensure the robustness of the findings.
In the multi-level nested/hierarchy there is client, provider (if one hospital is managed by multiple providers), hospital and perinatal unit. Authors have explained the reason for not including the provider in the model.
Is there any reason not to include hospital as one of the levels? (unless the hospital and provider are equivalent).
Though Appendix B clearly describes how they decided to choose between random intercept and random slope model bringing that small paragraph to the main manuscript will increase clarity of approach.

Validity of the findings

In the result section, authors have mentioned the response rate as 32%. But in the method, it was mentioned everyone participated and gave informed consent. This might need a clarification
If the response rate was 32% authors can’t claim for generalizability as mentioned in the discussion section unless they could ensure the responders and non-responders are similar in terms of age, education and self-rated health.

Additional comments

Statistically, this has been very well approached and vividly documented
However, while reading it gives a feeling there is more of statistical focus.
Adding quality improvement and challenges in assessing the quality in reproductive care, what has been conventionally followed, what are the lacunae of those approaches might add value to this article
In the result, section authors have mentioned the response rate as 32%. But in the method, it was mentioned everyone participated and gave informed consent. This might need a clarification
If the response rate was 32% authors can’t claim for generalizability as mentioned in the discussion section unless they could ensure the responders and non-responders are similar in terms of age, education and self-rated health.
Since we expect women who experienced complications may experience more of negative picture and complications are commonly occur during childbirth and postpartum attempt for sensitivity analysis will ensure the robustness of the findings.
In the multi-level nested/hierarchy there is client, provider (if one hospital is managed by multiple providers), hospital and perinatal unit. Authors have explained the reason for not including the provider in the model.
Is there any reason not to include hospital as one of the levels? (unless the hospital and provider are equivalent).
Though Appendix B clearly describes how they decided to choose between random intercept and random slope model bringing that small paragraph to the main manuscript will increase clarity of approach.

Reviewer 3 ·

Basic reporting

My compliments for a good written paper. Nevertheless, could you clarify the following thins:
line 79-80: 'Poor outcomes are infrequent, and specific low performance (for clients and units) into one direction easily averages out into other directions.'
Also, please use another word for 'true' (e.g. line 82). (Parts of) results of statistical analyses should not be associated with 'true'.
line 85: I believe 'should' should be removed from the sentence.
line 131: 'from several ongoing regional observational studies', what does this mean?
line 141: 'Since each hospital can be assigned, uniquely to one perinatal unit'. What about hospitals in urban areas (the 'randstad'? How did you assign these hospitals to perinatal units? Or did you remove those hospitals?
line 382: For the future, we recommend the use of ReproQ in maternity care.' For what purposes, the ReproQ may be used?

Experimental design

Thank you for providing a well defined study and good described design. I have one major comment. In line 335 you state that you avoided overfitting by limiting case-mix correction. However, how did you make sure that your corrected sufficiently for case-mix? Could you please elaborate on that (plus back up with the relevant statistics for your models)?

Validity of the findings

See comment in 2.

---

## Round 0.2 · accepted · Accept

The comments are well addressed

Reviewer 2 ·

Basic reporting

satisfactory

Experimental design

The appropriate design has been chosen to answer the research question

The comments given for the previous manuscript is well addressed.

Validity of the findings

The statistical approach and reporting of those methods were clearly documented and similar to the previous section suggested comments are addressed.

Additional comments

We appreciate authors efforts to address all the queries and rebuttal for each point.